# Infants recruit logic to learn about the social world

Nicolò Cesana-Arlotti [1✉], Ágnes Melinda Kovács[2] & Ernő Téglás[2]

When perceptually available information is scant, we can leverage logical connections among hypotheses to draw reliable conclusions that guide our reasoning and learning. We investigate whether this function of logical reasoning is present in infancy and aid understanding and learning about the social environment. In our task, infants watch reaching actions directed toward a hidden object whose identity is ambiguous between two alternatives and has to be inferred by elimination. Here we show that infants apply a disjunctive inference to identify the hidden object and use this logical conclusion to assess the consistency of the actions with a preference previously demonstrated by the agent and, importantly, also to acquire new knowledge regarding the preferences of the observed actor. These findings suggest that, early in life, preverbal logical reasoning functions as a reliable source of evidence that can support learning by offering a logical route for knowledge acquisition.

[1] Department of Psychological and Brain Sciences, The Johns Hopkins University, 3400 North Charles Street, Baltimore, MD 21218, USA. [2] Department of Cognitive Science, Cognitive Development Center, Central European University, 1051 Budapest, Hungary. ✉email: nicolocesanaarlotti@gmail.com

Logic can serve human cognition in a wide variety of ways. The compositionality of logical structures maximizes the flexibility of learning and reasoning[1–3], logical operations play a role in natural language syntax and the interpretation of sentences[4], while sensitivity to argument validity secures the correct evaluation of the reasons that we exchange[5]. Remarkably, however, logical reasoning can also be seen as a unique source of knowledge that can uncover truths outside the reach of other means, such as perception, memory, or communicative interactions. This epistemic function of logic seems invaluable for scientific progress, as it can enable seminal discoveries (e.g., proving by contradiction that the square root of two is an irrational number). More customarily, logical deduction also helps problem-solving by revealing facts that are concealed to other faculties (e.g., deducing the circumstances of an event that happened at a distant location or time-point). Importantly, logical conclusions offer a solid evidential basis for reasoning, learning, and decision-making processes, since they are as certain as the premises from which they are derived. As a consequence, the study of early logical abilities is essential for a better understanding of the cognitive tools that gift human infants with the potential of developing an adult-like mind[6]. Since logical reasoning can enhance knowledge acquisition by disclosing facts that are difficult or perhaps impossible to discover otherwise, a crucial question is whether logical deduction has this epistemic function when learning is much needed: in infancy and childhood.

We know surprisingly little regarding preverbal infants' logical abilities, if any exist at all. Traditionally, psychological research targeting logical reasoning has been most often motivated and informed by overt linguistic behavior or by the non-linguistic behavior of proficient users of a natural language[7,8]. Similarly, research on the development of logical abilities has mostly targeted the onset and the development of comprehension and production of the logical vocabulary: the words that can be used to express logical relations in natural languages (but see refs. [9,10]). The evidence collected so far suggests that basic logical words, like "no", "not" and "or" may be neither produced[11–17] nor understood[18–22] with logical meaning in the first two years of life. This lack of evidence for competence with logical language raises the questions of whether preverbal logical concepts exist, and if so, what function they might serve in the infant mind.

Recently, a few studies have started to investigate the presence of logical abilities before the acquisition of logical vocabulary, focusing on disjunctive reasoning. In a disjunctive inference, two or more alternative hypotheses are logically framed in a disjunctive relation (i.e., such that at least one of them must be true) and one of them is inferred because all the other alternatives are eliminated (e.g., either A or B; A is eliminated; therefore B). Thus, in its simplicity, this type of inference shows how logic allows for testing a hypothesis even when no evidence that directly confirms (or disconfirms) it is available.

In an experiment designed to produce stringent evidence for disjunctive reasoning[23], children younger than three years failed to logically update, compare and choose between two independent pairs of alternatives regarding the location of two hidden rewards (one in A or B, the other in C or D), to maximize the chance of retrieving one of them, after one alternative was eliminated (e.g., not A). The authors proposed two alternative interpretations of young children's failure. Children under the age of three years might be unable to represent disjunctive relations, possibly because they have not yet mastered the linguistic coordinator "or". Alternatively, even the younger children might be capable of performing disjunctive inferences, and their failure might reflect specific task-demands, that tax their less mature working memory capacity.

In contrast, a recent study[24] revealed that even infants may rely on early logical capacities to solve tasks that involve disjunctive reasoning. In this study, infants were presented with animated movies focusing on an object whose identity could not be unambiguously determined and was compatible with two alternative hypotheses. The joint use of measures of eye-movements, pupillary responses, and looking time at unexpected events provided multiple pieces of evidence that, when a disambiguating clue was available, infants spontaneously deployed disjunctive inference to derive the identity of the ambiguous object. Thus, while logical reasoning may still be developing during preschool years, at least the fundamental logical operations required for reasoning by elimination (of a disjunct) seem to predate the mastery of the logical vocabulary and possibly play a role in its acquisition.

However, the nature and function of preverbal logical inferences in the infant mind are still almost entirely uncharted. One could argue that infants may possess some proto-logical abilities, which resemble logical operations but are severely restricted in various aspects. For instance, infants' logical computations might be operational only in some restricted cognitive domains and thus might not fulfill similar functions in the infant mind as they do for adults. Crucially, it is an open question of whether early logical inferences can function as a source of evidence that efficiently supports infants' reasoning and learning about the physical and the social world. To play such an epistemic role, preverbal logical operations need to generate inferentially productive conclusions. That is, conclusions that can be integrated as input for other cognitive processes and thus result in chains of inferences. If preverbal logical conclusions are inferentially productive, infants' logical reasoning makes available data that are otherwise not accessible, thereby channeling the acquisition of new knowledge in a unique way.

Everyday examples reflecting the use of logical deductions as a source of evidence in social inferences are rather frequent. Imagine the following scene that requires drawing inferences about an agent's actions and goals. You watch a child in the park playing with a car and a ball on a bench. Suddenly, the child starts searching under the bench in the grass. When you notice that the ball is in fact in the child's lap, by elimination, you will think she must be searching for the car. Although apparently simple and effortless, such an explanation of the child's behavior rests on remarkably sophisticated reasoning feats: the outcome of a disjunctive inference has to be combined with the representation of an action to support an accurate interpretation of the child's intentions.

Inferring the intentions and preferences of social partners may constitute an ideal case study for investigating the productive contribution of preverbal logical abilities in knowledge acquisition. Differentiating incidental behavior from actions that are driven by goals and dispositions play an indispensable role in social learning and social interactions in infancy. Infants seem to be endowed with early abilities to understand others' goal-directed actions[25] which may already be present from the third month of life[26–28]. Crucially, however, even in its preverbal form, the representation of goal-directed behavior requires the integration of different kinds of information about the agent, the action, the physical constraints imposed by the environment, and the objects that are present in it. By considering the physical constraints imposed by the environment, infants can readily assess the efficiency of the actions and evaluate them as goal-directed or not[29–34]. Infants can also use further information about the circumstances of behavior to interpret a choice in terms of goals or object-directed dispositions. For example, when infants are presented with multiple repetitions of a scene where a hand grasped one of two potential goal-objects, they expect that

the agent will choose the same object again in new situations[35]. These results suggest that observing choices can serve as the basis for the attribution of a positive disposition toward the chosen object in early infancy[36,37].

While it is often the case that the factors relevant for understanding others' actions are perceptually available, in a large variety of cases they are not. In such cases, logical reasoning might serve as a useful tool for helping infants to interpret actions that would otherwise remain opaque and uninformative. In four experiments, we investigate infants' ability to use a disjunctive inference as a source of evidence in processing others' actions, that can aid social learning when more direct data is not available. First, we ask whether infants can integrate a logical conclusion with the representation of an ambiguous action to evaluate the consistency of its goal with a previously attributed disposition (Experiments 1–3). Most importantly, in Experiment 4, we ask whether infants' logical conclusions can empower social learning, by productively supporting the acquisition of new knowledge regarding the preferences of an agent.

## Results

**Experiment 1**. In Experiment 1, we investigate whether 14-month-old infants can infer via disjunctive reasoning the identity of an ambiguous object and use this logical conclusion to evaluate the consistency of an action with a previously encoded preference. First, infants were familiarized with animated movies where an agent repeatedly chose one of two fully visible objects (Supplementary Movies 1 and 2). Each trial began with the presentation of two centrally positioned objects (belonging to distinct familiar kinds: a toy car and a ball) that were then covered by two vertical occluders and moved toward the lower corners of the screen. Once the final position was reached, the occluders moved downward and revealed the upper fragments of the objects for a short time. Finally, the occluders disappeared, exposing the objects in full view, and a hand grasped one of the objects. This procedure was designed to familiarize infants with a critical feature of the two objects: specifically, that their top part was identical (Fig. 1a) so that they looked alike when their lower parts were covered. Across six familiarization trials, the agent always chose the same object (e.g., the car), henceforth, the goal-object, and never the competitor-object (e.g., the ball), the non-goal-object (objects counterbalanced across participants).

After familiarization, we tested infants' capacity to use disjunctive reasoning to evaluate the consistency of an ambiguous action using new movies (Supplementary Movies 3 and 4). In each test trial, infants were briefly presented with the same pair of objects used in familiarization. Then, the two objects were completely covered for a short time by two occluders, such that when their identical top part became visible, it was ambiguous which object was at a specific location. The occluders moved to the left and right side of the screen while the objects remained in partial occlusion. As a result, it was ambiguous which was the goal-object, but the scene could be captured as a disjunction of two alternatives (e.g., the goal-object is EITHER the object on the left OR the one on the right). Afterward, the location of the non-goal-object (i.e., the object never is chosen in familiarization) was briefly revealed: the object exited from behind the occluder and then returned to the partially hidden position (only the identical top part was visible. Fig. 1b). From this moment, all the evidence needed to deduce by the elimination of the location of the goal-object was available to the infants (i.e., since the revealed object is the non-goal-object, by elimination, the concealed object must be the goal-object). Then, a hand entered the scene and grasped the top of one of the two objects while they were still partially hidden. Across the four counterbalanced test trials, in two the hand

grasped the concealed object. This was the consistent choice since the concealed object was always the goal-object. In the other two trials, the hand grasped the previously revealed object. This was an inconsistent choice since the revealed object was always the non-goal-object.

Infants looked longer at the inconsistent choice than at the consistent one, even though they had no direct visual access to where the goal-object was located ($M_{consistent} = 6.9$ s, $M_{inconsistent} = 10$ s; $t(1, 23) = 2.5$, $P = 0.018$, $d = 0.51$; Fig. 2). This result is consistent with the hypothesis that infants inferred by elimination the identity of the concealed object (e.g., since the object on the left is the ball, by exclusion, the object on the right must be the car), and integrated such logical conclusion with the representation of actions directed toward the goal-object (e.g., she has chosen the car) or toward the other object. Such a logically enriched understanding of the actions helped infants to evaluate the consistency of the agent's choice with their previously acquired knowledge regarding her goals and dispositions.

However, one might argue that unlike in earlier studies targeting infants' goal understanding[35], in our test trials the non-goal-object might have become overly salient, as it was visually accessible, while it moved, pulsed, and emitted sounds just before the action of the agent (and this was not the case for the goal-object). Specifically, an alternative explanation for the observed looking patterns might be that infants allocated more attention (looked longer) to the action directed toward the more salient object. In Experiment 2 we aim to exclude the possibility that infants' looking pattern simply reflected such an asymmetry between the two test events and not their expectations of goal-directed actions based on logical reasoning.

Furthermore, one can think of yet another alternative explanation of the results of Experiment 1, proposing that infants could have detected an inconsistent choice without integrating a logical conclusion with the representation of an action. Specifically, during familiarization, infants might have noticed that the agent never chose the non-goal-object and thus attribute to the agent an avoidance goal or a negative disposition toward that object (e.g., she dislikes the ball; see ref. [38], for a discussion on the potential role of avoidance goals and negative dispositions). Crucially, the attribution of an avoidance goal might have supported the detection of the inconsistent choice without inferring by elimination the location of the goal-object. Since the location of the non-goal-object was always revealed in the test trials, no disjunctive reasoning might be required to see a conflict between having the intention of avoiding an object and yet choosing it. In Experiment 3, we aim to exclude this possibility by creating a situation in which the non-goal-object cannot have any role in the detection of the inconsistent choice.

**Experiment 2**. To exclude the possibility that infants' looking pattern in Experiment 1 simply reflected the higher saliency of the revealed object, in Experiment 2, we test whether infants look longer at the actions directed toward the revealed object even if they have no previous knowledge about the agent's preference for the other object. Several studies targeting goal attribution in infancy converged in showing that actions toward an object in isolation are often insufficient for encoding the identity of the goal-object[37] and for the attribution of a positive disposition toward it[33,36,39,40]. For example, when habituated with a hand grasping object A in isolation, infants typically do not look longer to a subsequent choice of B over A than to a choice of A over B. Building on these findings, in Experiment 2, we removed the competitor-object from the familiarization movies (Supplementary Movie 5). After the familiarization, infants were tested with the same test movies as in Experiment 1. Therefore, if infants'

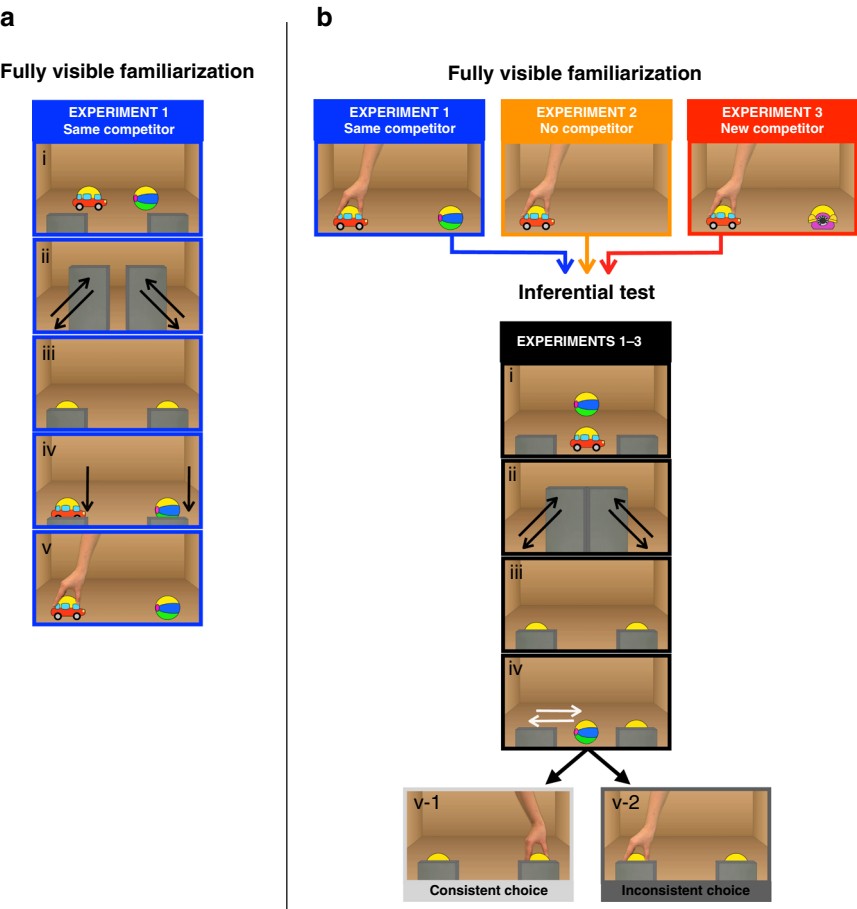

**Fig. 1 Experiments 1–3. Structure of the familiarization and the test movies. a** Familiarization events from Experiment 1, where infants saw hiding/ revealing events (i–iv) followed by actions toward a fully visible object (v). **b** The identical test events of Experiments 1–3 preceded by a different familiarization. Experiment 1: the objects used in familiarization were the same ones then used in the test (a toy car and a ball). Experiment 2: only the grasped object was present in familiarization (either the car or the ball). Experiment 3: the competitor-object (a toy telephone) used in familiarization was replaced in the test with a new object. In the test of Experiments 1–3, infants were presented with hiding events that prevented tracking the identity of the objects (i–iii). The location of the competitor-object was revealed; thus, infants could infer by eliminating the location of the familiar goal-object (iv). The hiding was followed by a consistent (v-1) or an inconsistent choice (v-2) targeting two partially covered objects.

looking pattern in Experiment 1 simply reflected the asymmetries between the test objects and not specific expectations of goal-directed actions derived from logical reasoning, we should expect the same pattern here as well.

However, in contrast to Experiment 1, in Experiment 2, infants looked longer at the consistent choice event, that is, when the hand reached to the concealed object ($M_{consistent} = 15.4$ s, $M_{inconsistent} = 6.6$ s; $t(1, 23) = 5.5$, $P = 0.0001$, $d = 1.12$; Fig. 2). This result excludes the possibility that infants looked longer at the inconsistent choice in Experiment 1 simply because its goal was more salient. A comparison of Experiments 1 and 2 revealed an interaction between choice type (consistent/inconsistent) and Experiment ($F(1, 46) = 35.3$, $P = 0.0001$), showing that when infants were not familiarized with the agent choosing one out of two objects, their looking pattern was different (Supplementary Note 1). This interaction suggests that encoding the goal of the agent in familiarization played a critical role in their surprise at the inconsistent test choice in Experiment 1. Thus, infants' performance in Experiment 1 most likely reflects that they successfully integrated the conclusion of a disjunctive inference with their representation of the agent's goals and actions.

It is noteworthy that, in Experiment 2, after familiarization with a reaching event involving a single object, infants looked longer in test at actions directed toward this object compared to a

new object. Such a pattern, however, was not reported in other studies using similar familiarization, where children were found to look equally at the two events[26,33,37,39]. Thus, this pattern is likely to reflect specific features of our procedure. Interestingly, this result is compatible with the hypothesis that, while in Experiment 1 infants learn about the agents' preference, in Experiment 2, in absence of a preference demonstration, participants may have formed the expectation that they will learn about objects. In contrast to previous studies, in the test phase of Experiment 2, the two reaching events were directed toward partially hidden objects and right before they reach, the new object was briefly revealed, while the familiar object remained concealed. Thus, without a familiarization with a preference between two objects, infants may have formed the expectation that something will be demonstrated about the object that moved out of occlusion. Consequently, they might have expected that the hand will reach for it and looked longer if this expectation was violated. This interpretation of Experiment 2 is, however, post-hoc and orthogonal to our questions, as our aim was solely to exclude an alternative explanation for Experiment 1.

**Experiment 3**. In Experiment 3, we ask whether 14-month-old infants could detect the inconsistent choice also when the

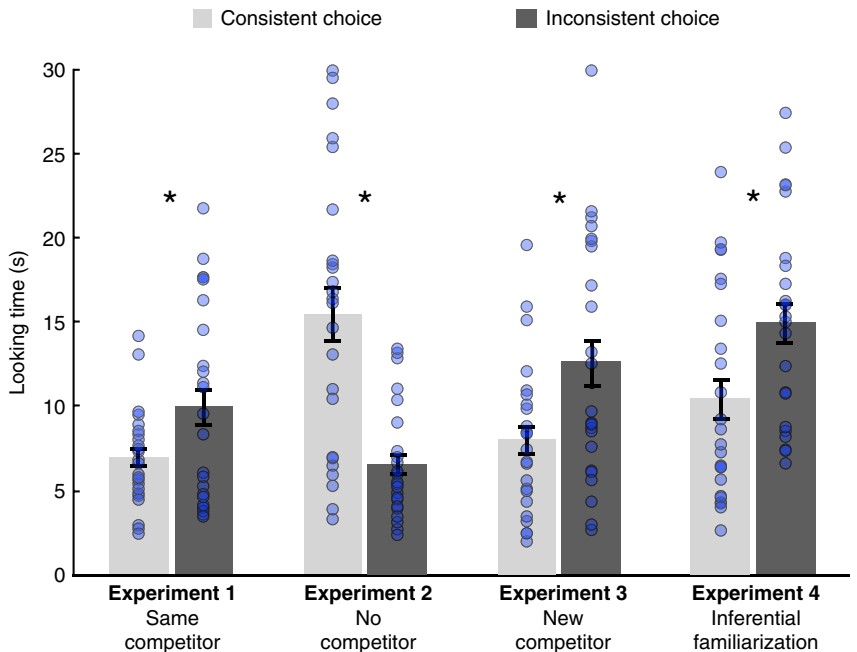

**Fig. 2 Infants looking times at the agent's choice in the test.** Mean (±SEM) looking times (in s) in the consistent (light gray) or inconsistent choice (dark gray) conditions in the four experiments, each involving an independent group of infants ($N = 24$, each). The blue dots show individual data points per participant for each experimental condition. Error bars represent the standard error of the mean. Asterisks indicate significant effects of the type of choice observed by the infants (Experiment 1: $t(1, 23) = 2.5$, $P = 0.018$; Experiment 2: $t(1, 23) = 5.5$, $P = 0.0001$; Experiment 3: $t(1, 23) = 2.8$, $P = 0.008$; Experiment 4: $t(1, 23) = 2.7$, $P = 0.011$; all tests were paired $t$-tests, two-tailed). Source data are provided as a Source Data file.

potential attribution of the intention to avoid the competitor-object in familiarization is of no help. Experiment 3 was identical to Experiment 1, except for one change: the competitor-object used in familiarization was replaced with a new object in the test (Supplementary Movie 6). Infants were familiarized to a choice between a pair of objects (e.g., choosing a toy car over a toy telephone) and then tested with choices between the chosen object and a new one (e.g., a toy car and a ball). While most of the studies targeting infants' goal attribution have focused on choices between the same set of alternatives, recent studies have shown that infants can generalize object-directed dispositions to contexts where the goal-object is contrasted with a new competitor[36–38]. That is, when infants are presented with multiple choices of an object A over an object B, they expect that the agent will choose object A over a new object C. Building on these results, we developed a test for the integration of logical conclusions with the representation of actions, where a potential attribution of an avoidance goal could not help the infants. Since in Experiment 3, the non-goal-object was replaced by a different object in the test, encoding intentions solely in terms of avoidance should prevent infants from forming expectations about the agent's actions at test. Therefore, if the attribution of a negative disposition was the only support for infants' detection of the inconsistent choice in Experiment 1, infants' looking time to the two test choices in the current experiment should not differ.

In Experiment 3, similarly to Experiment 1, infants looked longer at the inconsistent choice compared to the consistent one ($M_{consistent} = 8$ s, $M_{inconsistent} = 12.5$ s; $t(1, 23) = 2.8$, $P = 0.008$, $d = 0.58$; Fig. 2), indicating that they successfully detected the inconsistent action even when they had no knowledge about the agents' attitude toward the object revealed in the test. Thus, this result provides further evidence that infants identified the goal-object via disjunctive reasoning and integrated this conclusion with the representation of the ambiguous action. Taken together, the results of Experiments 1, 2, and 3 show that 14-month-old infants can integrate a logical conclusion with the representation

of an action to assess the consistency of its goal with a previously attributed disposition. More generally, these results indicate that preverbal logical conclusions offer an evidential basis to evaluate the consistency of observed social events (i.e., actions directed toward a concealed object) with previously acquired knowledge about the agent's preferences.

Such findings lead to the further critical question of whether the conclusion of a preverbal disjunctive inference can also function as evidence for the acquisition of completely new knowledge. In Experiment 4, we investigate the inferential productivity of preverbal logical deduction by asking whether infants' disjunctive inference can also support the encoding of new dispositions.

**Experiment 4**. In Experiment 4, we inquire if infants' logical conclusions regarding the identity of the hidden object could also support the attribution of a new preference. To test whether preverbal logical conclusions are a solid basis for such attribution we flipped the structure of the previous experiments, in the sense that the disjunctive inference was now necessary for encoding the goal in the familiarization phase (Fig. 3a). In the familiarization of Experiment 4, 14-month-old infants were acquainted with an ambiguous choice, where the concealed identity of the repeatedly chosen object was compatible with two possibilities and had to be disambiguated via elimination (Supplementary Movie 7). Afterward, we tested the successful attribution of a positive disposition toward the goal-object by showing choices between two fully visible objects, either of the goal-object (consistent choice) or of a new competitor (inconsistent choice, see Supplementary Movie 8). As in Experiment 3, we replaced the non-goal-object with a new one in the test to ensure that the detection of the inconsistent choice was not driven by the attribution of an avoidance goal (Fig. 3b). Thus, in Experiment 4, to detect the inconsistent test choice, infants had to attribute a new preference in familiarization based on evidence gained via disjunctive

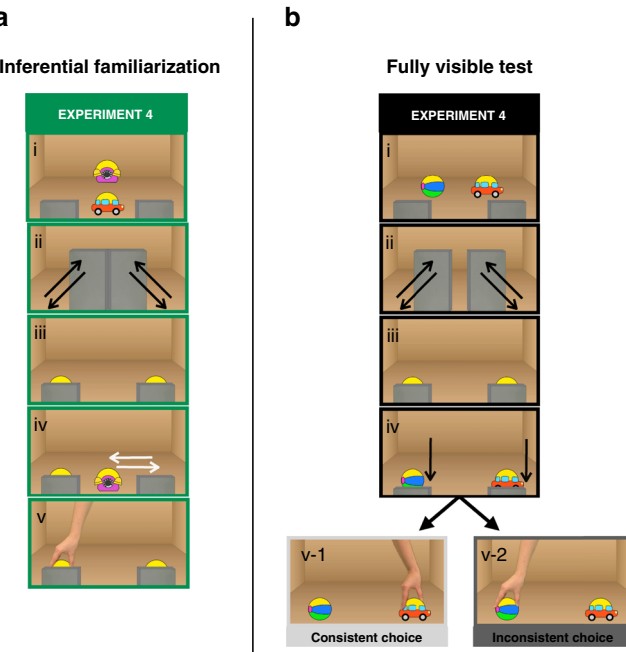

**a** Inferential familiarization

**b** Fully visible test

**Fig. 3 Experiment 4. Structure of the familiarization and test movies. a** Infants were familiarized with hiding events that prevented tracking the identity of the objects (i–iii), followed by actions toward a partially covered object (v), whose identity had to be inferred via disjunctive inference (iv). **b** Infants' knowledge of the agent's goal was tested with hiding/revealing events (i–iv) followed by a consistent (v-1) or an inconsistent choice (v-2) targeting two fully visible objects (the goal-object and a new competitor).

reasoning. That is, for infants to succeed in Experiment 4, their logical conclusion (e.g. the object on the right has to be the car, by exclusion) has to be integrated with the representation of an action (e.g., the agent chose the object on the right) and efficiently support the attribution of a preference (e.g., the agent must like the car), even though infants have never seen the agent grasp the object in full view. In contrast, if preverbal logical conclusions cannot function as an input for preference attribution then, this time, infants will not be able to form the appropriate expectations and look equally in the two conditions.

When presented with this new task, infants looked longer at the inconsistent choice ($M_{consistent} = 10.5$ s, $M_{inconsistent} = 14.9$ s; $t$ (1, 23) = 2.7, $P = 0.011$, $d = 0.56$; Fig. 2), similarly to Experiments 1 and 3. This result shows that infants' disjunctive deduction of the hidden identity of the object supported the recognition of the correct preference. Such knowledge of the agents' dispositions was readily formed and used to evaluate her subsequent choices in a new context. Remarkably, infants succeeded in encoding a preference based on a logical inference when acquainted with the same number of demonstrations as in the previous experiments where they had evidence based on direct perception. The similar performance in Experiments 3 and 4 (Supplementary Note 1) suggests that conclusions generated by preverbal disjunctive reasoning can function efficiently as a solid evidential ground for additional computations and, importantly, play a role in the acquisition of new knowledge.

## Discussion
While infants have access to a vast amount of information via perception or from their stored memories, some other, similarly important information can be outside the reach of these faculties. Logical reasoning can open a complementary route to facts that are not accessible from perception and memory alone: forming

expectations through logical deduction, as in the case of the elimination of alternatives. Hence, logical reasoning secures a way to knowledge acquisition by disclosing evidence otherwise not available. It was, however, an open empirical question whether this route to knowledge is available to preverbal human infants. In four experiments, we provide positive evidence to this query.

One might argue that the conclusions of preverbal disjunctive inferences could be severely limited in their integrability with other computational mechanisms. Thus, they might lack the inferential productivity required to assist reasoning and learning. Contrary to this possibility, results from Experiments 1–3 suggest that infants used logical conclusions disambiguating the hidden identity of an object to evaluate the consistency of an observed action with their background knowledge (i.e., their knowledge about the agent's dispositions). Crucially, in Experiment 4, infants successfully used the outcome of the disjunctive inference as a firm basis for the acquisition of new knowledge (i.e., the attribution of a preference to an agent) that supported expectations regarding future choices in a novel context. Remarkably, in this experiment infants were able to successfully encode the agent's disposition toward an object in a condition where the identity of the chosen object had to be inferred by elimination after the same number of demonstrations as in Experiment 3, where the chosen object was fully visible. Therefore, our results reveal that infants' logical reasoning is a powerful inferential device that has an interface already at a preverbal age with computational systems dedicated to distinct cognitive domains (e.g., the physical domain and the social domain[41]) and generates solid conclusions that efficiently function as evidence for reasoning and learning.

Importantly, our findings confirm that, at the beginning of their second year of life, infants are already capable of representing a disjunctive relation between two possible identities of a hidden object (i.e., at least one of the two has to be the correct one). Indeed, infants inferred the correct identity of the hidden object, without acquiring additional direct evidence in its support, but based on data that ruled out the other possibility. Although the inferential abilities hereby demonstrated by infants point at some form of preverbal disjunctive reasoning, they are compatible with multiple accounts regarding the format of the underlying logical representations. Infants' disjunctive inference might rest on explicit language-like mental operators linked to syntactic and inferential combinatorial rules (e.g., A OR B, NOT A, therefore B[8]). Alternatively, the disjunctive relation might be more implicitly captured by the representation of an exhaustive space of multiple mutually exclusive possibilities as models of the scene together with the process of updating it via elimination (e.g., {model A, model B}, model A is ruled out, therefore model B[7]). Importantly, further research is required to determine the nature of the process of elimination of alternatives used by infants in making disjunctive inferences. Is infants' discounting of one alternative—that leads to the logical confirmation of the remaining one(s)—a categorical process that fully rules it out or a graded operation that lowers its probability to values close to but different from 0? One might argue that a process of elimination based on a fine-graded weighting of negative evidence against one alternative might result in a richer contribution to the learning process. Independently of whether infants' update of alternatives is categorical or probabilistic, to successfully infer the correct identity of the hidden object in our task infants has to represent its potential identities in a logical relation of disjunction—i.e., at least one of the alternatives must be correct, thus evidence against one confirms the other.

In a recent paper, Leahy and Carey have suggested that children younger than 4 years may lack the ability to represent multiple mutually exclusive alternatives and perform computations to update them, and they may approximate logical

expectations via serial guessing[42]. While it is unclear how such guessing could take place without representing the space of the alternatives (which would be equivalent to implementing a disjunctive relation), suppose that, just like in our experiments, there are two objects hidden in two possible locations and infants first randomly guess the identity of a hidden object. Lacking the logical prerequisites of updating the priors based on a disjunctive relation between alternatives, when infants see evidence that is inconsistent with their first guess, they formulate a new random guess with the same priors. Importantly, in this example, such sequentially performed guessing may result in a correct solution only 75% of the time. Applying this procedure across the familiarization trials of Experiment 4 the distribution of incorrect/ correct solutions should be 1:3. However, previous research has demonstrated that preference attribution is disrupted when an agent is seen making inconsistent choices (in one-fourth of the familiarization trials the agent chooses object A, while in the rest object B[43]). Thus, infants' success in Experiment 4 of our study is unlikely to be explained by relying on simple serial guessing. Instead, these findings are in line with the possibility that a preverbal form of disjunctive inference may be in place early on. Importantly, as also highlighted by Leahy and Carey, with the accumulating evidence and the contribution of future studies we hope to gain a better understanding regarding the nature of the computations underlying logical abilities in infants and young children.

The current results also open new questions and motivate further empirical investigations regarding the nature and scope of logical computations in the preverbal mind. First, while the present finding indicates that a representation of a disjunctive relation is a logical primitive in the infant mind, future research should try to identify the potential precursors of the other concepts that are central to human logical reasoning. Are preverbal forms of logical quantification (e.g., the relations expressed by words such as "each" and "all"[44]) and modal reasoning (e.g., the inferences grounded in the representation of possibility and necessity[45]) available early in life?

Second, recent research suggests that infants are also able to perform transitive inferences involving dominance and preference relations[46–48]. An especially interesting venue for future research is comparing the preverbal precursors of disjunctive and transitive reasoning and asking whether the two inferences use distinct kinds of representation (e.g., logical rules or mental models) or else they employ the same representational system, but rely on different relations. For instance, while transitive inferences are often suggested to be based on a mental model of a linear order (see[49,50] for findings with children and adults), disjunctive inferences likely involve not a single model, but a space of multiple models of mutually exclusive possibilities[51].

Third, infants' success in combining a logical conclusion about the identity of an object with the representation of an action to attribute a preference, suggests that preverbal logical inferences documented here may be similar to those of adults in that they can support the integration of information among distinct cognitive domains. This integrability raises the question of whether infants' representation of disjunction can be deployed to reason about various domains of experience distinct from physical objecthood, similarly to the domain-independent logical operators of natural language.

Furthermore, data providing evidence for the epistemic role of logical inferences in human infants also raise a critical question for comparative studies. Numerous experiments have revealed problem-solving performance in many non-human species that are consistent with disjunctive reasoning (see ref. [52] for a review). However, some authors have argued for the need for more evidence regarding the presence of non-human logical

representations[23,53]. Besides novel tests targeting non-human disjunctive abilities (see for an example, ref. [54]), it may become of primary interest to ask whether potential logical inferences are a source of evidence for learning in a non-human mind. Is the ability to use logical reasoning to efficiently gain knowledge about the dispositions of conspecifics, and other factors of the environment supporting vital predictions, specific to humans, or is it shared with other, non-human species?

While future investigations should address different emerging questions regarding the nature, scope, and phylogenies of such logical abilities, the present finding provides evidence for a powerful inferential capacity that may support human infants' outstanding learning performance: preverbal logical reasoning. When perceptual evidence is scant, infants can draw conclusions via the elimination of alternatives. Preverbal logical conclusions are integrable representations and compelling evidence that motivate infants to draw new inferences, like those involved in reasoning about actions and goals. Thus, infants' capacity to generate logical expectations, and exploit them in subsequent computations, may greatly increase their learning opportunities. Consequently, preverbal logical reasoning may offer to the infant learner a unique route to knowledge.

## Methods

**Participants.** A total of 96 healthy full-term 14-month-old infants were included in the analysis: Experiment 1 ($N = 24$; $M_{age} = 14$ m 02d, range 13 m 15d–14 m 14d; 16 girls), Experiment 2 ($N = 24$; $M_{age} = 14$ m 02d, range 13 m 16d–14 m 16d; 10 girls), Experiment 3 ($N = 24$; $M_{age} = 14$ m 06d, range 13 m 19d–14 m 15d; 13 girls), Experiment 4 ($N = 24$; $M_{age} = 13$ m 29d, range 13 m 14d–14 m 11d; 9 girls). The parents gave informed consent for participation in the research. We have complied with all relevant ethical regulations. The study was approved by the United Ethical Review Committee for Research in Psychology (EPKEB) in Hungary. (For information about recruitment and inclusion criteria see Supplementary Note 2).

## Materials

*Experiment 1.* Each infant watched 6 familiarization movies and 4 test movies. The movies were presented on a 24-inch screen with PsyScope X, which controlled the experiment, running on an Apple Mac Mini 2,8 GHz Intel Core i5. In the familiarization phase, participants saw movies where, trial after trial, a hand always grasped the same object out of two (a ball and a toy car). In each movie, the two objects were initially briefly introduced and positioned horizontally near each other. Then, they were fully covered by two separates vertical occluders. Afterward, the top halves of the objects became uncovered, showing that their top parts were identical. Then, the occluders moved with their object, one to the left side of the screen and the other to the right side, while the objects remained in partial occlusion. Afterward, the two objects were simultaneously fully uncovered. For the entire time, the two occluders were clearly spatially separated, diminishing in this way the possibility for errors in object tracking. At this point, the hand entered the scene from the upper edge, briefly stopped, and then grasped the top of one of the two objects. At the moment of the reaching, the two objects were fully visible. (For movies details, timing and counterbalancing of all experiments see Supplementary Note 3). In Experiment 1 infants were familiarized with a choice between the same pair of objects that were also used in the test (Supplementary Movies 1 and 2). In the test phase, participants saw movies where, trial after trial, the hand grasped one out of two partially covered objects, while the location of the goal-object (i.e., the object repeatedly is chosen in familiarization) had to be inferred via a disjunctive inference. In each movie, first, the two objects were briefly introduced and positioned vertically one above the other. Then, the two objects were completely covered for a short time by two occluders, such that when their identical top part became visible, it was ambiguous which object was at a specific location. Afterward, the top half of the objects became uncovered. Then, the occluders moved with their object, one to the left side of the screen and the other to the right side, while the objects remained in partial occlusion. Afterward, the location of the non-goal-object was briefly revealed: the object exited from behind the occluder, pulsed and emitted a sound, and then returned to the partially hidden position. At this point, similarly to the familiarization a hand entered the scene and grasped the top of one of the two objects. Differently from familiarization, the objects were still partially covered. In two test movies, the hand grasped the non-revealed goal-object (consistent choice), in the other two the revealed non-goal-object (inconsistent choice) (Supplementary Movies 3 and 4). In order to draw participants' attention to the events of the movie (e.g., object introduction, object displacement, and reappearing of the competitor-object in the test) these were accompanied by sounds both in familiarization and in the test. At the start of each trial and after the appearance of the agent's hand, right before the grasping action, a female voice was played ('Hello baby, hello' and 'Look at this') to call the attention of the infants.

**Experiment 2.** We changed the material of Experiment 1 by removing the competitor-object from the familiarization (Supplementary Movie 5). Thus, in Experiment 2, half of the participants were familiarized with a hand reaching for a ball in the absence of any competitor and the other half with a hand reaching for a car in the absence of any competitor.

**Experiment 3.** The only change with respect to Experiment 1 was that in familiarization the competitor-object was replaced by a toy telephone (Supplementary Movie 6) and served as a competitor in all movies. Thus, in Experiment 3, half of the participants were familiarized with a choice of a ball over a telephone and the other half with the choice of a car over a telephone. However, in the test phase a different competitor-object was featured (a car or a ball, respectively). Importantly, the test movies were exactly the same as the ones used in Experiments 1 and 2.

**Experiment 4.** The materials of Experiment 4 were identical to those of Experiment 3 except for the following changes. The structure of the test movies and of the familiarization movies was switched. The familiarization procedure was designed to acquaint infants with an agent's object-directed disposition when the identity of the chosen object had to be derived via logical reasoning. The familiarization movies of Experiment 4 were identical to the test movies of the previous Experiments 1–3, except that a toy telephone was used as a competitor (Supplementary Movie 7). Crucially, in each familiarization trial, the identity of the chosen object had to be inferred by elimination. Half of the participants were familiarized with the ball as the goal-object while the other half with the car as the goal-object. For all participants, the non-goal-object was always the telephone. The test movies of Experiment 4 were identical to the familiarization movies of Experiment 1. Thus, at the moment of reaching, the two objects were both fully visible.

### Procedure

**Experiments 1–4.** The experiment took place in a sound-proof room with dimmed lights. Participants were seated on their caregiver's lap, at about 60 cm distance from the display. The caregivers wore opaque glasses that prevented them from seeing the stimuli. They were instructed to keep the child seated on their laps and not to interact with them. The experimenter was seated behind a curtain and monitored infants' behavior, from a separate screen via a video camera. Infants first watched six familiarization trials that had a fixed duration and displayed a scene where the hand always grasped the same object. The last scene of the reaching was displayed for an additional 4 s to ensure that infants have encoded the reach. Afterward, infants were exposed to four test trials. In each test trial, at the end of each movie, the last frame remained on, and looking times were recorded till one of the below criteria was reached. A test trial ended when participants looked away for at least 2 s or when they looked at the screen for a total of 30 s. The entire session was video-recorded. To extract exact looking time durations for data analysis looking behavior was coded offline with PsyCode. (For additional information see Supplementary Note 4).

**Analyses.** For each experiment, we ran a preliminary two-way repeated-measures ANOVA with the type of the choice (consistent choice/inconsistent choice) as a within-participant factor and goal-object type (Ball/Car) as a between-participants factor. The analyses revealed a main effect of the type of choice and no other main effects or interactions (Supplementary Note 1). Therefore, we collapsed looking times along the object type, and compared infants' average-looking times in the consistent and inconsistent choice conditions with a paired two-tailed t-test, as reported in the main text. Cohen's d estimates of effect size for the paired $t$-test comparisons were calculated based on the Mean and Standard Deviation of the difference between average-looking times in the two conditions. Data were analyzed with the DataDesk 8 Statistical Analysis software (https://datadescription.com).

**Reporting summary.** Further information on research design is available in the Nature Research Reporting Summary linked to this article.

### Data availability

The data that support the findings of this study are available in the OSF repository, http://osf.io/adbfs. The source data underlying Fig. 2 is provided as a Source Data file. A reporting summary for this Article is available as a Supplementary Information file. Source data are provided with this paper.

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

## Acknowledgements

We thank parents and children participating in this study and K. Andrási, Z. Karap, M. Nagy, M. Toth, and A. Volein for their help with data collection and K. Begus, L. Bonatti, G. Csibra, L. Feigenson, and J. Halberda for their invaluable comments on the manuscript. This study was supported by a European Union's Seventh Framework Programme (FP7/2007-2013) ERC Grant 284236 REPCOLLAB and by a European Union's Horizon 2020 Research and Innovation Programme ERC Grant 639840 PreLog.

## Author contributions

N.C.-A., Á.M.K., and E.T. designed the study; N.C.-A. performed the research; N.C.-A. and E.T. analyzed the data; and N.C.-A., Á.M.K., and E.T. wrote the paper.

## Competing interests

The authors declare no competing interests.
