## [Peer Review File · Nature Communications]

Reviewers' comments:

Reviewer #1 (Remarks to the Author):

This paper presents four experiments that explore 14-month-old infants' ability to make disjunctive inferences (either A or B, so if Not A, then B) and use these inferences to reason about an agent's goal-directed actions. Experiments 1 and 3 demonstrate that when observing an agent select between two ambiguous partially-occluded, infants used disjunctive inference to determine the identity of the selected object and looked longer when the agent's choice was inconsistent with a previously established preference. In Experiment 4, infants did the reverse: in a series of familiarization trials in which an agent selected between two ambiguous partially-occluded objects, infants used disjunctive inference to identify which object she selected and thereby infer her preference, then use this to evaluate the consistency of her choice in later test trials with fully visible objects.

The manuscript is clear and well written, the experiments are clever and well designed, and the results and conclusions seem sound. The findings are striking and make important contributions to our understanding both of infants logical abilities and their early social understanding – in particular, the results of Experiment 4 show that infants can attribute to an agent a preference for an object, even when the identity of that object is not yet known, and I believe this may be the first demonstration of this in the literature.

I have only minor suggestions for the author to consider:

- The results of Experiment 1 alone cannot prove infants used disjunctive inference because infants could simply have looked longer when the agent selected the non-preferred object. This alternative is fully addressed by the results of Experiment 3. However, it would be helpful to acknowledge this issue earlier in the discussion of the results of Experiment 1. Otherwise it appears that the authors are overstating the claims from that experiment. It would be sufficient to simply acknowledge both alternatives (salience, avoidance) at the end of Experiment 1 and then test those in the two subsequent experiments.

- The SI describes the counterbalancing for each familiarization and test phase. However, it does not explain how the two were combined – i.e. whether the last familiarization reach and the first test reach were on the same side 50% of the time, etc. Given the detailed counterbalancing, I suspect this was the case. However, it would be helpful to clarify this, both to address possible low-level criticisms and to facilitate replication.

- It is unclear whether some or all trials had a minimum looking time. In the inclusion criteria section, it states that participants were excluded if they looked at an outcome for less than 2s. Is this specific to the infant-controlled test trials? Or does it apply to familiarization as well? This criterion is also not mentioned in the procedure section of the main manuscript, which only states the look away and maximum time. If there was a minimum looking time, it should be listed there as well.

Reviewer #2 (Remarks to the Author):

The authors report in four experiments that 14-month-old infants can use disjunctive syllogisms to make inferences about an agent's goals and preferences. It is a novel finding and should be of interest to other areas of research. I enjoyed reading the manuscript and had a few suggestions for revision.

First, there have been at least a couple of studies reporting infants' understanding of transitive inferences, which is also considered logical thought (if $A > B$ and $B > C$, then $A > C$) (Gazes, Hampton, & Lourenco, 2017; Mou, Province, & Luo, 2014). The authors may want to discuss how "special" disjunctive syllogisms are, e.g., their links to language.

Second, I am surprised by the results of Experiment 2, because as the authors pointed out, previous studies usually report null test results when choice information is lacking in similar familiarization trials, presumably because when a new object is paired with the familiar object in tests, both visible, infants have no predictions as to which one the agent should choose. But in Experiment 2, the infants still might be induced to infer the location of the familiar object during test and make predictions that the agent should choose the new object, which is puzzling. I wonder if the authors could discuss more on this. For example, would similar results obtain if the familiarization trials were eliminated altogether?

Finally, the familiarization movies were of slightly different lengths in the four experiments. It seems that the authors did not measure infants' looking times during familiarization. I wonder whether it is possible to code off the video records how long infants looked in the six familiarization trials across experiments. These familiarization looking times are usually reported in studies on infants' goal attribution abilities. One of the reasons is that it could ensure that before test trials infants have been similarly attentive and hence make cross-experiment comparisons more convincing.

Reviewer #3 (Remarks to the Author):

This paper reports four infant experiments investigating whether they can use disjunctive reasoning in thinking about an agent's preferences. The methods are very clever, combining elements of Cesena-Arlotti et al. (2018) and Woodward (1998). The results are very interesting and compelling.

I think of this series of experiments as following the lead of the Cesena-Arlotti et al. (2018) study. I am unsure if the findings are general enough for Nature or perhaps they should be published in a more specialized journal.

Two comments:

(1) Looking time data are positively skewed, so in addition to the simple t-tests, the authors may want to

include non-parametric tests as well.

(2) I also have one substantive comment: How do we know this is really logical reasoning? That is, one could argue that when one of the two objects left the scene, the infant lowered the probability of that object to almost 0 and raised the probability of the other object to nearly 1. That is, the looking time method may only give us information about how probable or improbable an outcome is, since all we can measure is relative looking times to the two outcomes: consistent vs. inconsistent – longer looking to the inconsistent outcome tells us that this is the less probable outcome than the consistent outcome. If this interpretation of the method is correct, we cannot conclude that infants eliminated one of the objects in favor of the other object. I think this line of thought applies to the earlier Cesana-Arlotti et al. (2018) study as well. Again, the experiments are very interesting, but I want to raise the above conceptual question for the authors to ponder.

RESPONSE TO REVIEWER #1.

REVIEWER #1:

This paper presents four experiments that explore 14-month-old infants' ability to make disjunctive inferences (either A or B, so if Not A, then B) and use these inferences to reason about an agent's goal-directed actions. Experiments 1 and 3 demonstrate that when observing an agent select between two ambiguous partially-occluded, infants used disjunctive inference to determine the identity of the selected object and looked longer when the agent's choice was inconsistent with a previously established preference. In Experiment 4, infants did the reverse: in a series of familiarization trials in which an agent selected between two ambiguous partially-occluded objects, infants used disjunctive inference to identify which object she selected and thereby infer her preference, then use this to evaluate the consistency of her choice in later test trials with fully visible objects.

The manuscript is clear and well written, the experiments are clever and well designed, and the results and conclusions seem sound. The findings are striking and make important contributions to our understanding both of infants' logical abilities and their early social understanding – in particular, the results of Experiment 4 show that infants can attribute to an agent a preference for an object, even when the identity of that object is not yet known, and I believe this may be the first demonstration of this in the literature.

We are happy that Reviewer 1 finds our results striking and important, and we thank Reviewer 1 for the detailed suggestions on how to improve the manuscript.

The results of Experiment 1 alone cannot prove infants used disjunctive inference because infants could simply have looked longer when the agent selected the non-preferred object. This alternative is fully addressed by the results of Experiment 3. However, it would be helpful to acknowledge this issue earlier in the discussion of the results of Experiment 1. Otherwise it appears that the authors are overstating the claims from that experiment. It would be sufficient to simply acknowledge both alternatives (salience, avoidance) at the end of Experiment 1 and then test those in the two subsequent experiments.

As the reviewer pointed out, Experiments 2 and 3 address two alternative interpretations of Experiment 1. In the original manuscript we have acknowledged both those two alternatives, but only one at the time, in the discussion of Experiment 1 and of Experiment 2, respectively. Following the reviewer's suggestion, we now describe both alternatives in the discussion of the results of Experiment 1 (main text, page 6, 4th paragraph). Then we address the two alternatives in the discussion of Experiment 2 and 3, respectively.

The SI describes the counterbalancing for each familiarization and test phase. However, it does not explain how the two were combined – i.e. whether the last

familiarization reach and the first test reach were on the same side 50% of the time, etc. Given the detailed counterbalancing, I suspect this was the case. However, it would be helpful to clarify this, both to address possible low-level criticisms and to facilitate replication.

The familiarization and test phase were combined exactly as the reviewer implies: in each of the four experiments, the side of the chose object in the last familiarization trial and in the first test trial was the same for half of the participants, and the opposite for the other half. We now describe this feature of our experimental design more carefully in the SI (SI, page 3, 1st paragraph).

It is unclear whether some or all trials had a minimum looking time. In the inclusion criteria section, it states that participants were excluded if they looked at an outcome for less than 2s. Is this specific to the infant-controlled test trials? Or does it apply to familiarization as well? This criterion is also not mentioned in the procedure section of the main manuscript, which only states the look away and maximum time. If there was a minimum looking time, it should be listed there as well.

We thank the reviewer for pointing out this potential ambiguity in the submitted manuscript. In familiarization, each infant was presented with six trials that had a fixed duration, and were not designed for measuring looking times (but see new information about how much attention infants have paid during the familiarization trials in the revised manuscript; SI, page 5, 3rd paragraph). In contrast, in the infant-controlled test trials, we used a criterion for trial inclusion of at least 2 seconds looking at the end of the trial to ensure sure that infants had processed the test choice (Buttelmann & Kovacs, 2019, p744). We have described this in the inclusion criteria section and not in the procedure section, because it was applied during the offline coding of the looking times. We rephrased the description of the familiarization procedure (main text, page 16, 2nd paragraph). Also, the description of the exclusion criterion in the SI was rephrased to make it clearer (SI, page 3, 2nd paragraph).

Buttelmann, F., & Kovács, Á. M. (2019). 14- Month- olds anticipate others' actions based on their belief about an object's identity. *Infancy*, 24(5), 738–751.
<https://doi.org/10.1111/infa.12303>

RESPONSE TO REVIEWER #2.

REVIEWER #2:

The authors report in four experiments that 14-month-old infants can use disjunctive syllogisms to make inferences about an agent's goals and preferences. It is a novel finding and should be of interest to other areas of research. I enjoyed reading the manuscript and had a few suggestions for revision.

We are thankful for the reviewer's insightful questions and suggestions about how to improve the manuscript. We are also happy to read that the reviewer enjoyed reading our manuscript.

First, there have been at least a couple of studies reporting infants' understanding of transitive inferences, which is also considered logical thought (if $A > B$ and $B > C$, then $A > C$) (Gazes, Hampton, & Lourenco, 2017; Mou, Province, & Luo, 2014). The authors may want to discuss how "special" disjunctive syllogisms are, e.g., their links to language.

As the reviewer pointed out, transitive inferences of the form (i) "A is dominant over B; B is dominant over C; therefore, A is dominant over C" seem to be also available to preverbal infants (Gazes, Hampton, & Lourenco, 2017; and see also Mou, Province, & Luo, 2014 regarding the transitivity of preferences). Whether infants' transitive and disjunctive inferences are based on similar processes is an exciting issue. We think that this question deserves to be pursued in the future.

While a comprehensive analysis of the differences between transitive and disjunctive inferences present in philosophical and psychological theories goes beyond the focus of the present manuscript, below we highlight a couple of relevant considerations that reflect important features of disjunctive inferences that we have also discussed in our manuscript.

Disjunctive inferences are based on representations of truth-functional relations (i.e., disjunction) that are abstract, in the sense that carry no assumptions regarding the structure of the domains to which they are applied (a point suggested by Tarski, 1986). Given this feature, disjunctive inferences can be an especially useful tool for flexible hypothesis testing (we have briefly addressed this point in the main text, introduction page. 1). In contrast, transitive inferences seem to require specific assumptions about the relations involved. For instance, to reason transitively about dominance (as in Gazes, Hampton, & Lourenco, 2017) requires assuming that the dominance relations are asymmetrical and transitive. While such assumption of transitivity might be correct in many cases, it might not hold in every case (see the example in the next paragraph).

Most relevant for studying their role in infants' knowledge acquisition, disjunctive inferences are valid in the strong sense that it is impossible for the premises of the inference to be true and the conclusion to be false. Importantly, it is this strong validity of disjunctive inferences that may make them an especially reliable source of evidence for supporting learning and reasoning (a function that we discussed in the main text,

introduction, page 2). However, in contrast, it is unclear whether transitive inferences based on dominance and preference are valid in this strong sense. For example, it can be true that 'The general is dominant over the major' and that 'The major is dominant over her German Shepherd', while it is false that 'The general is dominant over the major's German Shepherd'.

Two important psychological theories of logical reasoning, the mental logic (e.g., Braine & O'Brien, 1998; Rips, 1994) and the mental model theory (Johnson-Laird, 1995), provide competing alternative accounts regarding the processes involved in disjunctive inferences. Our results are consistent with either of these two accounts, and each of them highlight (distinct) reasons why studying disjunctive reasoning in infancy may be of special interest. However, both seem to imply that disjunctive inferences are different from transitive ones.

According to the mental logic theory, logical reasoning is guided by logical rules based on concepts of truth-functional operators (OR, NOT, AND, IF... THEN) and quantifiers (ALL, MOST, SOME). However, it was argued that children's and adults' transitive inferences observed in various studies may not be based on logical rules, but instead on a model of a linearly ordered relation (Riley and Trabasso, 1974, Goodwin and Johnson-Laird, 2008). The above-mentioned infant studies also adhere to such an explanation. Thus, for the mental logic theory, studying infants' disjunctive inferences might be of special interest as they are suitable candidates to provide evidence for preverbal logical rules (as we mentioned in the discussion, page 12).

According to the mental model theory of logical reasoning, disjunctive inferences rely on the representation of multiple models of mutually exclusive possibilities – corresponding to the alternatives in the relation of disjunction –, while this is not the case for transitive inferences (Mackiewicz & Johnson-Laird, 2011). Thus, based on the mental model theory, preverbal disjunctive inferences might be diagnostic of infants' capacity of simultaneously representing multiple mutually exclusive possibilities (as we mentioned in the discussion, page 12).

We have added to the revised manuscript a paragraph about infants' transitive inferences and their potential relation to disjunctive inferences (main text, page 12, 3rd paragraph; reported below):

“Second, recent research suggest that infants are also able to perform transitive inferences involving dominance and preference relations⁴⁴⁻⁴⁶. An especially interesting venue for future research is comparing the preverbal precursors of disjunctive and transitive reasoning and asking whether the two inferences use distinct kinds of representation (e.g., logical rules or mental models) or else they employ the same representational system, but rely on different relations. For instance, while transitive inferences are often suggested to be based on a mental model of a linear order (see^{47,48} for findings with children and adults), disjunctive inferences

likely involve not a single model, but a space of multiple models of mutually exclusive possibilities⁴⁹.”

Second, I am surprised by the results of Experiment 2, because as the authors pointed out, previous studies usually report null test results when choice information is lacking in similar familiarization trials, presumably because when a new object is paired with the familiar object in tests, both visible, infants have no predictions as to which one the agent should choose. But in Experiment 2, the infants still might be induced to infer the location of the familiar object during test and make predictions that the agent should choose the new object, which is puzzling. I wonder if the authors could discuss more on this. For example, would similar results obtain if the familiarization trials were eliminated altogether?

Experiment 2 was designed to exclude an alternative interpretation of Experiment 1: that looking time patterns observed in Experiment 1 reflect an increased attention for the actions directed toward the competitor object because of the object's higher saliency in test. As the reviewer pointed out, in Experiment 2, we found a very different pattern compared to Experiment 1, which excludes this alternative interpretation. However, we did not find a null-result as in other studies where preference attribution was eliminated from familiarization in the very same way (see Biro et al., 2011; Baillargeon et al., 2015; Robson and Kuhlmeier, 2019; Luo and Baillargeon, 2005; Luo, 2011), but a significant difference in the opposite direction compared to Experiment 1.

While our hypothesis predicted the interaction between Experiment 1 and 2 (main text, page 7), it remained agnostic regarding the specific pattern of Experiment 2 (besides being different from Experiment 1). Thus, given that we predicted only the interaction to exclude the alternative explanation, in the earlier version of our manuscript we did not include post-hoc explanations regarding the pattern observed in Experiment 2. However, motivated by the reviewer's question here we provide one possible explanation, which should be considered with due caution:

The pattern of Experiment 2 (and that of Experiment 1) could be related to the different inferences that infants may have made regarding to *what they are demonstrated about* in different contexts. Note that all our movies included ostensive communication (infant directed speech) and were followed by actions that could have been interpreted by infants as demonstrations guiding their attention in specific ways, as suggested by the natural pedagogy theory (Csibra and Gergely 2009). However, the different contexts instantiated in Experiment 1 and 2 (i.e., the preferential choice between two objects in Experiment 1 vs. no choice in Experiment 2) might have induced different expectations regarding what the demonstrations were about.

Specifically, in Experiment 1 where a choice between two objects was demonstrated in familiarization, infants likely *learned about (the preference of) the agent*. However, in Experiment 2 in the absence of a choice demonstration in familiarization, they might infer that the demonstration *is about (the properties of) objects*. Given that the structure

of the test highlights the novel object (which moves out of occlusion before the reach), infants may form the expectation that something will be demonstrated about that object, and thus expect the hand to reach for it and show surprise if this expectation is violated.

As raised by the reviewer, it is unclear whether the results we found in Experiment 2 would persist if the familiarization trials were eliminated altogether. However, it is possible that the removal of the familiarization might not yield exactly the same results, since processing the test events – without any familiarization with the elements of the scene – might be more demanding for the infants and hinder their understanding of the scene. Importantly, in designing Experiment 2 we followed previous studies in adopting a single object familiarization, rather than eliminating the familiarization altogether. This manipulation has the advantage of making the comparison between Experiment 1 and 2 more warranted, since in both Experiments infants were familiarized to the same number of reaching event toward the same object, with the only difference being the presence/absence of a competitor.

While we would like to be cautious with the interpretation of the looking pattern in Experiment 2 that is orthogonal to our hypothesis, we have updated the discussion of the result of Experiment 2 to include some of these considerations (main text, page 8, 2nd paragraph; reported below):

“It is noteworthy that, In Experiment 2, after familiarization with a reaching event involving a single object, infants looked longer in test at actions directed toward this object compared to a new object. Such a pattern, however, was not reported in other studies using similar familiarizations, where children were found to look equally at the two events^{39,33,26,37}. Thus, this pattern is likely to reflect specific features of our procedure. Interestingly, this result is compatible with the hypothesis that, while in Experiment 1 infants learn about the agents’ preference, in Experiment 2 in absence of a preference demonstration participants may have formed the expectation that they will learn about objects. In contrast to previous studies, in the test phase of Experiment 2, the two reaching events were directed toward partially hidden objects and right before the reach the new object was briefly revealed, while the familiar object remained concealed. Thus, without a familiarization with a preference between two objects, infants may have formed the expectation that something will be demonstrated about the object that moved out of occlusion. Consequently, they might have expected that the hand will reach for it and looked longer if this expectation was violated. This interpretation of Experiment 2 is, however, post-hoc and orthogonal to our questions, as our aim was solely to exclude an alternative explanation for Experiment 1.”

Finally, the familiarization movies were of slightly different lengths in the four experiments. It seems that the authors did not measure infants' looking times during familiarization. I wonder whether it is possible to code off the video records how long infants looked in the six familiarization trials across experiments. These familiarization looking times are usually reported in studies on infants' goal attribution abilities. One of the reasons is that it could ensure that before test trials infants have been similarly attentive and hence make cross-experiment comparisons more convincing.

Unlike the test trials, the familiarization trials had a fixed duration and we did not measure looking times after the presentation of the stimuli. We did not analyze infants' attention toward the familiarization stimuli, since the same reaching event (with minor variations – e.g. the side) was repeated six times and thus we believed that our procedure offers infants plenty of evidence to encode the reach. Additionally, in all experiments, infants found the familiarization movies highly attractive and paid careful attention to them. However, pondering the question of the reviewer, we agree that an exact quantitative description of infants' behavior *during* the familiarization phase may be informative.

To check infants' attention *during* the familiarization of each experiment, following the reviewer's suggestion, we have now coded the total viewing time (the time spent looking at the stimuli) during each familiarization trial for each of the four experiments (570 trials in total).

We have included in the SI an analysis that compares these looking proportion across experiments (see SI, page 5, 3rd paragraph; reported below):

“To check infants' attention during the familiarization of each experiment we measured the total *viewing time* (the time spent looking at the stimuli) during each familiarization trial for each of the four experiments. Given the variation across experiments of the familiarization trial length, we have calculated the proportion of time that the stimuli were attended by a participant (total viewing time divided by trial length). Crucially, in each experiment infants were highly attentive to the familiarization stimuli (mean proportions: $M_{\text{Experiment1}} = .97$, $M_{\text{Experiment2}} = .96$, $M_{\text{Experiment3}} = .97$, $M_{\text{Experiment4}} = .93$). To compare infants' attention during familiarization between experiments, we run one-way repeated measures ANOVA with experiment (Experiment1/ Experiment2/ Experiment3/ Experiment4) as a between-participants factor comparing the average proportion of time that infants have spent viewing the familiarization movies. One participant from Experiment 3 was not included in this analysis because of incomplete data due to a technical error. The analysis yielded a main effect of experiment ($F(3, 91) = 4.46$, $P = 0.005$). Infants' average viewing proportion was

lower in Experiment 4 than in Experiment 1 (Scheffè test, $P = 0.016$) and then in Experiment 3 (Scheffè test, $P = 0.035$), with no other differences. The slightly lower viewing proportion in Experiment 4 is likely due to the fact that familiarization trials in Experiment 4 were longer compared to the familiarization trials in Experiment 1 and 3. In any case, infants demonstrated the same level of interest toward the familiarization stimuli of Experiment 1 and 2, suggesting that the interaction observed in the response to the test movies of these experiments cannot be explained with a difference in infants' attention to the familiarization stimuli.”

Baillargeon, R., Scott, R. M., He, Z., Sloane, S., Setoh, P., Jin, K., Wu, D., & Bian, L. (2015). Psychological and sociomoral reasoning in infancy. In M. Mikulincer, P. R. Shaver, E. Borgida, & J. A. Bargh (Eds.), *APA handbook of personality and social psychology, Volume 1: Attitudes and social cognition*. (pp. 79–150). American Psychological Association.
<https://doi.org/10.1037/14341-003>

Biro, S., Verschoor, S., & Coenen, L. (2011). Evidence for a unitary goal concept in 12-month-old infants: Unitary goal concept in 12-month-olds. *Developmental Science*, *14*(6), 1255–1260.
<https://doi.org/10.1111/j.1467-7687.2011.01042.x>

Braine, M. D. S., & O'Brien, D. P. (Eds.). (1998). *Mental logic*. L. Erlbaum Associates.

Csibra, G., & Gergely, G. (2009). Natural pedagogy. *Trends in Cognitive Sciences*, *13*(4), 148–153.
<https://doi.org/10.1016/j.tics.2009.01.005>

Gazes, R. P., Hampton, R. R., & Lourenco, S. F. (2017). Transitive inference of social dominance by human infants. *Developmental Science*, *20*(2), e12367. <https://doi.org/10.1111/desc.12367>

Goodwin, G. P., & Johnson-Laird, P. N. (2008). Transitive and pseudo-transitive inferences. *Cognition*, *108*(2), 320–352. <https://doi.org/10.1016/j.cognition.2008.02.010>

Johnson-Laird, P. N. (1995). *Mental models: Towards a cognitive science of language, inference, and consciousness* (6. print). Harvard Univ. Press.

Luo, Y., & Baillargeon, R. (2005). Can a Self-Propelled Box Have a Goal?: Psychological Reasoning in 5-Month-Old Infants. *Psychological Science*, *16*(8), 601–608.
<https://doi.org/10.1111/j.1467-9280.2005.01582.x>

- Luo, Yuyan. (2011). Three-month-old infants attribute goals to a non-human agent: Three-month-olds attribute goals to a non-human agent. *Developmental Science*, *14*(2), 453–460. <https://doi.org/10.1111/j.1467-7687.2010.00995.x>
- Mackiewicz, R., & Johnson-Laird, P. N. (2012). Reasoning from connectives and relations between entities. *Memory & Cognition*, *40*(2), 266–279. <https://doi.org/10.3758/s13421-011-0150-8>
- Mou, Y., Province, J. M., & Luo, Y. (2014). Can infants make transitive inferences? *Cognitive Psychology*, *68*, 98–112. <https://doi.org/10.1016/j.cogpsych.2013.11.003>
- Riley, C. A., & Trabasso, T. (1974). Comparatives, logical structures, and encoding in a transitive inference task. *Journal of Experimental Child Psychology*, *17*(2), 187–203. [https://doi.org/10.1016/0022-0965\(74\)90065-4](https://doi.org/10.1016/0022-0965(74)90065-4)
- Rips, L. J. (1994). *The psychology of proof: Deductive reasoning in human thinking*. MIT Press.
- Robson, S., & Kuhlmeier, V. A. (2019). *Reaching for an object or that object: Context and limitations in encoding object features may constrain infant goal attribution* [Preprint]. PsyArXiv. <https://doi.org/10.31234/osf.io/hj7f9>
- Tarski, A., & Corcoran, J. (1986). What are logical notions? *History and Philosophy of Logic*, *7*(2), 143–154. <https://doi.org/10.1080/01445348608837096>

RESPONSE TO REVIEWER #3.

This paper reports four infant experiments investigating whether they can use disjunctive reasoning in thinking about an agent's preferences. The methods are very clever, combining elements of Cesena-Arlotti et al. (2018) and Woodward (1998). The results are very interesting and compelling.

We are happy that the reviewer liked our study and expressed interest in our findings.

I think of this series of experiments as following the lead of the Cesena-Arlotti et al. (2018) study. I am unsure if the findings are general enough for Nature or perhaps they should be published in a more specialized journal.

Two comments:

(1) Looking time data are positively skewed, so in addition to the simple t-tests, the authors may want to include non-parametric tests as well.

As the reviewer suggested, we have added the results of non-parametric tests in the SI (SI, page 4, 5th paragraph), that converge with the parametric tests:

(2) I also have one substantive comment: How do we know this is really logical reasoning? That is, one could argue that when one of the two objects left the scene, the infant lowered the probability of that object to almost 0 and raised the probability of the other object to nearly 1. That is, the looking time method may only give us information about how probable or improbable an outcome is, since all we can measure is relative looking times to the two outcomes: consistent vs. inconsistent – longer looking to the inconsistent outcome tells us that this is the less probable outcome than the consistent outcome.

If this interpretation of the method is correct, we cannot conclude that infants eliminated one of the objects in favor of the other object. I think this line of thought applies to the earlier Cesena-Arlotti et al. (2018) study as well. Again, the experiments are very interesting, but I want to raise the above conceptual question for the authors to ponder.

We thank the reviewer for raising this interesting point about the potential probabilistic character of infants' expectations and its implications for the study of logical representations in the infant mind.

We agree with the reviewer that the proposal that looking times in Violation of Expectation (VoE) paradigms might reflect probabilistic expectations is a very interesting assumption. This interpretation of looking times can be used to make fine-grained predictions about responses to outcomes with different probabilities. In fact,

such an approach was successfully applied in one of our earlier works (Teglas et al., 2011). Furthermore, this proposal accommodates well probabilistic computational models that specify ways to use logical structures (and other abstract representations) to reason and learn from uncertain data (Goodman et al., 2015; Tenenbaum et al., 2011). However, such an interpretation of looking times is not implied by the VoE paradigm per se, but reflects a particular view of infants' cognition that should be targeted by further empirical research.

Most importantly, regarding the theoretical argument of the reviewer, while it might be indeed the case that, when infants are presented with evidence inconsistent with one alternative, the probabilities do not drop to exactly zero or go up to exactly one, this feature is actually not required for the thinking to be logical. Our study targets the processes by which infants update their expectations - that we argue are guided by logical inferences - and not whether the involved representations are probabilistic or categorical. For instance, when in our task infants see a ball coming out from behind the occluder on the left side (i.e. Figure 1B, iv; notice that in our movies no object leaves the scene), to succeed by updating probabilities, they have to lower the probability that the hidden object behind the right occluder is the ball close to 0. Crucially, they also have to rise the probability that the hidden object behind the right occluder is the car close to 1. This last update assumes that the probability of the second alternative is negatively correlated with the probability of the first alternative. Such negative correlation of probabilities is a consequence of the two alternatives being in a logical relation of disjunction (i.e., at least one of the expectations must be correct, so the less probable is one, the most probable is the other). Therefore, infants' disjunctive reasoning would be likely logical, independently of whether the representations they are operating with are probabilistic rather than categorical. We now introduce in the discussion part the point of whether elimination in preverbal disjunctive reasoning is categorical or probabilistic (main text, page 12, 1st paragraph; reported below):

“Importantly, further research is required to determine the nature of the process of elimination of alternatives used by infants in making disjunctive inferences. Is infants' discounting of one alternative - that leads to the logical confirmation of the remaining one(s) - a categorical process that fully rules it out or a graded operation that lowers its probability to values close to but different from 0? One might argue that a process of elimination based on a fine-graded weighting of negative evidence against one alternative might result in a richer contribution to the learning process. Independently of whether infants' update of alternatives is categorical or probabilistic, to successfully infer the correct identity of the hidden object in our task infants have to represent its potential identities in a logical relation of disjunction – i.e., at least one of the alternatives must be correct, thus evidence against one confirms the other.”

Thus, while the reviewer has raised a very interesting question, it does not change our conclusion that infants' use disjunctive reasoning as a source of evidence for further inferences in the social domain.

Goodman, N. D., Tenenbaum, J. B., & Gerstenberg, T. (2015). Concepts in a Probabilistic Language of Thought. In E. Margolis & S. Laurence (Eds.), *The conceptual mind: New directions in the study of concepts* (pp. 623–654). MIT Press.

Teglas, E., Vul, E., Girotto, V., Gonzalez, M., Tenenbaum, J. B., & Bonatti, L. L. (2011). Pure Reasoning in 12-Month-Old Infants as Probabilistic Inference. *Science*, *332*(6033), 1054–1059. <https://doi.org/10.1126/science.1196404>

Tenenbaum, J. B., Kemp, C., Griffiths, T. L., & Goodman, N. D. (2011). How to Grow a Mind: Statistics, Structure, and Abstraction. *Science*, *331*(6022), 1279–1285. <https://doi.org/10.1126/science.1192788>

REVIEWER COMMENTS

Reviewer #1 (Remarks to the Author):

I appreciate the thoughtful, detailed responses the authors have provided to the previous round of reviews. The changes to the manuscript/SI have strengthened and clarified the paper. All of my previous comments have been fully addressed. I have no further suggestions for the authors.

Reviewer #2 (Remarks to the Author):

The authors have addressed all my comments from the first round of review. However, I see now that the Wilcoxon Sign-ranks test of Experiment 1 was only marginally significant. It is a bit worrisome that only about half, 13 to be exact, of the 24 infants in Experiment 1 looked longer at Inconsistent than at the Consistent test event, $p = .067$ (on the data sheet, the consistent and inconsistent seemed reversed for Expt4). I would like the authors to comment on this.

Reviewer #3 (Remarks to the Author):

The authors have revised the manuscript based on the reviewers' comments. Recently Leahy and Carey (2020, Trends in Cognitive Sciences) have provided a critique of infant research on logical representations, and they argued that so far there is no convincing evidence yet. I find their critique compelling, and I too worry about infant researchers being too eager to attribute abilities to young learners, so I urge the authors to take a look at this new article.

RESPONSE TO REVIEWER #1.

REVIEWER #1:

I appreciate the thoughtful, detailed responses the authors have provided to the previous round of reviews. The changes to the manuscript/SI have strengthened and clarified the paper. All of my previous comments have been fully addressed. I have no further suggestions for the authors.

We are very happy that we have managed to address all the raised issues and would like to thank the reviewer again for the insightful comments and the suggestions on how to improve our manuscript. The paper has gained in strength and clarity based on the reviewer's feedback.

RESPONSE TO REVIEWER #2.

REVIEWER #2:

The authors have addressed all my comments from the first round of review. However, I see now that the Wilcoxon Sign-ranks test of Experiment 1 was only 'marginally significant. It is a bit worrisome that only about half, 13 to be exact, of the 24 infants in Experiment 1 looked longer at Inconsistent than at the Consistent test event, $p = .067$ (on the data sheet, the consistent and inconsistent seemed reversed for Expt4). I would like the authors to comment on this.

We are glad to read that the reviewer agrees that we have addressed all the raised questions and suggestions. We thank the reviewer for the insightful comments, and for giving us the opportunity to enrich the discussion of the results and to clarify the implications of our conclusions relative to the existing literature on the development of reasoning in infancy.

As the reviewer noticed, in the Source Data file the consistent and inconsistent conditions of Experiment 4 were reversed. We thank very much to the reviewer for pointing this out, we have corrected this.

The reviewer has asked us to comment on the nature of the data in Experiment 1. Specifically, while as reported in the manuscript, on page 6, the average looking times were higher in the inconsistent condition than in the consistent one, at the level of individual infants, there seems to be some variability in the samples. However, such individual variability is not uncommon in infant research, and, the typical sample size used in infant experiments results in some limitations when non-parametric analysis is applied. While the non-parametric tests strongly converge with the parametric analysis across the four experiments (Experiment 1: t-test: $t(1, 23) = 2.5$, $P = 0.018$; Wilcoxon test: $Z = 1.8$, $P = 0.067$; Experiment 2: t-test: $t(1, 23) = 5.5$, $P = 0.0001$; Wilcoxon test: $Z = -3.9$, $P = 0.0001$; Experiment 3: t-test: $t(1, 23) = 2.8$, $P = 0.008$; Wilcoxon test: $Z = 2.4$, $P = 0.015$; Experiment 4: t-test: $t(1, 23) = 2.7$, $P = 0.011$; Wilcoxon test: $Z = 2.8$, $P = 0.005$, all tests were two-tailed) in Experiment 1 they likely reflect such variability.

Importantly, our conclusions are based on the data pattern obtained in four experiments, which consist of one control study (Experiment 2) and two successful conceptual replications of the pattern observed in Experiment 1 (Experiments 3 and 4). Thus, we believe that the initial finding of Experiment 1, together with data from three consecutive experiments, provide strong support for the proposal we are making.

In the revised version of the manuscript we discuss the issue concerning the variability of looking patterns and clarify the assessment of the overall pattern of results (SI, page 4):

“In addition, we run Wilcoxon signed-ranks tests to examine individual infants' average looking times in the consistent and inconsistent choice conditions. This analysis

converges with the main analysis; in Experiments 1, 3 and 4 infants looked longer at the inconsistent choice than at the consistent one (Experiment 1: $Z = 1.8$, $P = 0.067$; Experiment 3: $Z = 2.4$, $P = 0.015$; Experiment 4: $Z = 2.8$, $P = 0.005$, all tests were two-tailed), while in the control Experiment 2 they looked longer at the consistent choice than at the inconsistent one (Experiment 2: $Z = -3.9$, $P = 0.0001$, two-tailed). The non-parametric analysis in Experiment 1 likely reflects individual variability that is not uncommon in infant research, and a reduced sensitivity of these tests at our sample size. Importantly, our conclusions are based on the data pattern obtained in four experiments, which consist of one control study (Experiment 2) and two successful conceptual replications of the pattern observed in Experiment 1 (Experiments 3 and 4). Thus, we believe that the initial finding, together with the two conceptual replications, provide a strong support for the proposal we are making aiming to enrich our understanding of infants' logical abilities.”

RESPONSE TO REVIEWER #3.

The authors have revised the manuscript based on the reviewers' comments. Recently Leahy and Carey (2020, Trends in Cognitive Sciences) have provided a critique of infant research on logical representations, and they argued that so far there is no convincing evidence yet. I find their critique compelling, and I too worry about infant researchers being too eager to attribute abilities to young learners, so I urge the authors to take a look at this new article.

We thank the reviewer for pointing out a paper by Leahy and Carey published in January 2020. We believe that the present study (specifically Experiment 4) offers evidence that may speak to some of the issues raised by Leahy and Carey.

In their analysis centered around the performance of preschool-aged children in specific studies (e.g. Mody & Carey, 2016; Redshaw & Suddendorf, 2016), Leahy and Carey have arrived at the conclusion that young children's understanding of ambiguous or uncertain events might be limited. According to the authors, infants and young children are not able to represent a space of two or more alternatives (which would be equivalent to implementing a disjunction) nor to update the space of alternatives when there is evidence inconsistent with one of them. Instead, they have suggested that younger children and infants might *guess* randomly among alternatives.

One aspect of the proposal of Leahy and Carey that we find puzzling is how one could guess with a 50 % probability of success between A and B without representing a space of mutually exclusive alternatives. Without representing a specific set of alternatives, guessing may involve a whole range of other options and result in outcomes that are irrelevant or even impossible. In fact, computational studies targeting guessing (see for example Bonawitz 2014) argue that guessing presupposes a space of alternatives. Thus, it seems to us that the guessing account proposed by Leahy and Carey presupposes exactly the kind of logical repertoire the authors are arguing against.

However, putting aside these unclear aspects of Leahy and Carey's arguments, we will now consider their proposal with respect to the present research findings. Leahy and Carey suggested that even in absence of the capacity of updating the space of alternatives, serial guessing in some cases could loosely approximate the outcome of a logical inference at the population level. To illustrate this idea, one would have to assume that in our study infants have two relevant time points where they could formulate a guess. For instance, when they are presented with two hidden objects at the beginning of the trial, they would randomly make a first guess with 50% probability of success (i.e., in half of the trials they will guess that the object hidden on the right is the car, while in the other half that it is the ball). Thus, when the identity of one of the two objects is revealed (Figure 1B), this event will be consistent with their first guess in half of the trials, but inconsistent in the remaining trials. When the evidence is consistent, the infants will maintain their correct guesses (50%). When it is inconsistent (50%), they will try to guess again. Crucially, if infants are unable to represent and update the space of alternatives (as suggested by Leahy and Carey), their second guess will have the

same probabilities (priors) as the first one (leading to success in 50% of these trials). Therefore, after two guesses, infants will end up with a correct guess in 75% of the trials.

Our current experiments, and especially Experiment 4, offers an opportunity to contrast this model of non-logical serial guessing with disjunctive inferences where an update of the space of alternatives takes place.

Specifically, during the familiarization of Experiment 4, infants have to logically infer (or else guess) what object has been chosen by the agent, since the chosen object is occluded at the time of the reaching. The serial guessing hypothesis (unlike disjunctive reasoning) predicts that, across the six familiarization trials, in 75% of the trials infants will compute that the agent has chosen the car (correct outcome) while in the remaining 25% that she has chosen the ball (incorrect outcome).

However, it is unlikely that infants were using such guessing in Experiment 4 of our study, given that previous research has demonstrated that preference attribution is disrupted if an agent is seen making inconsistent choices (e.g., Luo et al., 2017, where the agent chooses object A in one fourth of the familiarization trials, while object B in the rest). Given that data from Experiment 4 shows that infants have successfully attributed a preference to the agent, infants in our experiments have likely used a disjunction to infer the exact identity of the hidden object and to attribute a consistent preference to the agent.

We hope that our study will motivate further research in this direction and together with the evidence accumulated by future research will contribute in further advancing our knowledge regarding the nature of the computations underlying disjunctive reasoning in infants and in young children in different tasks.

To address the reviewer's comment, we have now added the following paragraph to the discussion (Main Text, page 12):

“In a recent paper Leahy and Carey (2020) have suggested that children younger than 4 years may lack the ability to represent multiple mutually exclusive alternatives and perform computations to update them, and they may approximate logical expectations via serial

guessing. While it is unclear how such guessing could take place without representing the space of the alternatives (which would be equivalent to implementing a disjunctive relation), suppose that, just like in our experiments, there are two objects hidden in two possible locations and infants first randomly guess the identity of a hidden object. Lacking the logical prerequisites of updating the priors based on a disjunctive relation between alternatives, when infants see evidence that is inconsistent with their first guess, they formulate a new random guess with the same priors. Importantly, in this example, such sequentially performed guessing may result in a correct solution only 75% of the time. Applying this procedure across the familiarization trials of Experiment 4 the distribution of incorrect/correct solutions should be 1:3. However, previous research has demonstrated that preference attribution is disrupted when an agent is seen making inconsistent choices (in one fourth of the familiarization trials the agent chooses object A, while in the rest object B, Luo et al., 2017). Thus, infants' success in Experiment 4 of our study is unlikely to be explained by relying on simple serial guessing. Instead, these findings are in line with the possibility that a preverbal form of disjunctive inference may be in place early on. Importantly, as also highlighted by Leahy and Carey (2020), with the accumulating evidence and the contribution of future studies we hope to gain a better understanding regarding the nature of the computations underlying logical abilities in infants and young children.”

Leahy, B. P. & Carey, S. E. The Acquisition of Modal Concepts. *Trends in Cognitive Sciences* **24**, 65–78 (2020).

Bonawitz, E., Denison, S., Gopnik, A. & Griffiths, T. L. Win-Stay, Lose-Sample: A simple sequential algorithm for approximating Bayesian inference. *Cognitive Psychology* **74**, 35–65 (2014).

Luo, Y., Hennefield, L., Mou, Y., vanMarle, K. & Markson, L. Infants' Understanding of Preferences When Agents Make Inconsistent Choices. *Infancy* **22**, 843–856 (2017).